# Mapping genes for human face shape: Exploration of univariate phenotyping strategies

Meng Yuan[1,2,3]*, Seppe Goovaerts[2,3]*, Michiel Vanneste[2,3], Harold Matthews[2,3,4], Hanne Hoskens[1,2,3,5], Stephen Richmond[6], Ophir D. Klein[7,8], Richard A. Spritz[9], Benedikt Hallgrimsson[5], Susan Walsh[10], Mark D. Shriver[11], John R. Shaffer[12,13], Seth M. Weinberg[12,13,14], Hilde Peeters[2], Peter Claes[1,2,3,4]*

1 Department of Electrical Engineering, ESAT/PSI, KU Leuven, Leuven, Belgium, 2 Department of Human Genetics, KU Leuven, Leuven, Belgium, 3 Medical Imaging Research Center, University Hospitals Leuven, Leuven, Belgium, 4 Murdoch Children's Research Institute, Melbourne, Victoria, Australia, 5 Department of Cell Biology & Anatomy, Cumming School of Medicine, Alberta Children's Hospital Research Institute, University of Calgary, Calgary, Alberta, Canada, 6 Applied Clinical Research and Public Health, School of Dentistry, Cardiff University, Cardiff, United Kingdom, 7 Departments of Orofacial Sciences and Pediatrics, and Institute for Human Genetics, University of California, San Francisco, San Francisco, California, United States of America, 8 Department of Pediatrics, Cedars-Sinai Guerin Children's, Los Angeles, California, United States of America, 9 Human Medical Genetics and Genomics Program, University of Colorado School of Medicine, Aurora, Colorado, United States of America, 10 Department of Biology, Indiana University Indianapolis, Indianapolis, Indiana, United States of America, 11 Department of Anthropology, Pennsylvania State University, State College, Pennsylvania, United States of America, 12 Center for Craniofacial and Dental Genetics, Department of Oral and Craniofacial Sciences, University of Pittsburgh, Pittsburgh, Pennsylvania, United States of America, 13 Department of Human Genetics, University of Pittsburgh, Pittsburgh, Pennsylvania, United States of America, 14 Department of Anthropology, University of Pittsburgh, Pittsburgh, Pennsylvania, United States of America

☯ These authors contributed equally to this work.

* meng.yuan@kuleuven.be (MY); seppe.goovaerts@kuleuven.be (SG); peter.claes@kuleuven.be (PC)

**Data Availability Statement:** The genotype data of the 3DFN dataset are accessible via the dbGaP controlled access repository (http://www.ncbi.nlm.nih.gov/gap) at accession number phs000949.v1.

## Abstract

Human facial shape, while strongly heritable, involves both genetic and structural complexity, necessitating precise phenotyping for accurate assessment. Common phenotyping strategies include simplifying 3D facial features into univariate traits such as anthropometric measurements (e.g., inter-landmark distances), unsupervised dimensionality reductions (e.g., principal component analysis (PCA) and auto-encoder (AE) approaches), and assessing resemblance to particular facial gestalts (e.g., syndromic facial archetypes). This study provides a comparative assessment of these strategies in genome-wide association studies (GWASs) of 3D facial shape. Specifically, we investigated inter-landmark distances, PCA and AE-derived latent dimensions, and facial resemblance to random, extreme, and syndromic gestalts within a GWAS of 8,426 individuals of recent European ancestry. Inter-landmark distances exhibit the highest SNP-based heritability as estimated via LD score regression, followed by AE dimensions. Conversely, resemblance scores to extreme and syndromic facial gestalts display the lowest heritability, in line with expectations. Notably, the aggregation of multiple GWASs on facial resemblance to random gestalts reveals the highest number of independent genetic loci. This novel, easy-to-implement phenotyping approach holds significant promise for capturing genetically relevant morphological traits

p1. The phenotype data, represented as 3D facial surface in .obj format, are available through the FaceBase Consortium (https://www.facebase.org) at accession number FB00000491.01. Access to these 3D facial surface models requires proper institutional ethics approval and approval from the FaceBase data access committee. The FaceBase repository in the syndromic face database, "Developing 3D Craniofacial Morphometry Data and Tools to Transform Dysmorphology", collected at patient support groups in the USA, Canada, and the UK. Facial images are available through FaceBase (https://www.facebase.org/chaise/record/#1/isa:dataset/accession=FB00000861). The participants making up the Peter Hammond's legacy 3D dysmorphology dataset, Penn State University (PSU) and Indiana University Indianapolis (IUI) datasets were not collected with broad data sharing consent. Given the highly identifiable nature of both facial and genomic information and unresolved issues regarding risks to participants of reidentification, participants were not consented for inclusion in public repositories or the posting of individual data. This restriction is not because of any personal or commercial interests. Further information about access to the raw 3D facial images and/or genomic data can be obtained from the respective ethics committees; the Ethics Committee Research UZ/KU Leuven (ec@uzleuven.be), the PSU IRB (IRB-ORP@psu.edu), and the IUI IRB (irb@iu.edu) for the Peter Hammond's legacy, PSU and IUI datasets, respectively. For the ALSPAC (UK) data, please note that the study website contains details of all the data that is available through a fully searchable data dictionary and variable search tool (http://www.bristol.ac.uk/alspac/researchers/our-data/). Genome wide genotyping data was generated by Sample Logistics and Genotyping Facilities at Welcome Sanger Institute and LabCorp (Laboratory Corporation of America) using support from 23andMe. All relevant source data for future replications are provided online (https://doi.org/10.6084/m9.figshare.24867063). This includes: the facial template, nasal landmark labels, the mesh simplification scheme used in AE models, the list of genetic loci associated with the nose and face shape, the GO biological processes based on the union set of lead SNPs from all groups of phenotypes, and the LocusZoom plots for each significant SNP based on different phenotyping methods. An example of LocusZoom plot can be found in Fig G in S1 File. Code availability KU Leuven provides the MeshMonk v.0.0.6 spatially dense facial-mapping software, free to use for academic purposes (https://github.com/TheWebMonks/meshmonk). MATLAB R2017b

derived from complex biomedical imaging datasets, and its applications extend beyond faces. Nevertheless, these different phenotyping strategies capture different genetic influences on craniofacial shape. Thus, it remains valuable to explore these strategies individually and in combination to gain a more comprehensive understanding of the genetic factors underlying craniofacial shape and related traits.

## Author summary

Advancements linking variation in the human genome to phenotypes have rapidly evolved in recent decades and have revealed that most human traits are influenced by genetic variants to at least some degree. While many traits, such as stature, are straightforward to acquire and investigate, the multivariate and multipartite nature of facial shape makes quantification more challenging. In this study, we compared the impact of different facial phenotyping approaches on gene mapping outcomes. Our findings suggest that the choice of facial phenotyping method has an impact on apparent trait heritability and the ability to detect genetic association signals. These results offer valuable insights into the importance of phenotyping in genetic investigations, especially when dealing with highly complex morphological traits.

## Introduction

Human facial development is highly complex, resulting in a rich diversity of facial appearances both within and among populations. Furthermore, facial features have a strong genetic basis, readily apparent within families. The genome-wide association scan (GWAS) is an agnostic approach designed to investigate the statistical relationship between phenotypic traits and genetic variants. A typical GWAS involves individually testing millions of single nucleotide polymorphisms (SNPs) or other common variants dispersed across the genome. Because the precise location of SNPs and genes is known, GWAS signals showing strong evidence of association can point to genes of interest. While many human traits are relatively straightforward to acquire, capturing facial variation is considerably less so, due to the multivariate and multipartite nature of faces.

Since the initial two GWASs on components of typical-range facial shape variation in 2012 [1,2], more than 300 genome-wide significant signals have been identified in over 20 different studies [3]. Several recent studies [4–8] have embraced a multivariate GWAS framework, regressing multiple univariate traits simultaneously onto each SNP genotype, and have thereby outperformed univariate GWAS in terms of genetic discovery. Nevertheless, several compelling arguments favor univariate GWAS. First, univariate GWAS results can be easily combined across studies via meta-analysis, thereby enhancing statistical power while obviating the need to share highly sensitive facial and genomic data. Second, several important follow-up analyses and GWAS applications, such as linkage disequilibrium score regression (LDSC) [9] and polygenic risk score calculations, require signed effect size and error estimates, which are not readily provided by multivariate techniques. Finally, univariate GWAS is simpler to execute and demands fewer computational resources than multivariate GWAS.

In a traditional anthropometric approach to facial phenotyping, researchers collect a set of univariate measurements such as the distances between pairs of well recognizable, sparsely distributed facial landmarks [1,2,10–18]. Newer approaches have used geometric morphometrics

implementations of the hierarchical spectral clustering to obtain nasal segmentation are available from a previous publication (https://doi.org/10.6084/m9.figshare.7649024). Code for training AE models is available at https://github.com/mm-yuan/autoencoder_3dface. The analyses in this work were based on functions in MALAB R2022b, Python v3.7.8, MeshMonk v0.0.6, MeshLab v2020.03, LDSC v.1.0.1, GREAT v4.0.4.

**Funding:** The KU Leuven research team (P.C., M.Y., S.G.) and analyses were supported by the Research Fund KU Leuven (BOF-C1, C14/20/081), and the Research Foundation-Flanders (FWO, G0D1923N). This work was funded in part by grants from the National Institute of Dental and Craniofacial Research: R01-DE027023 (S.M.W., J.R.S., P.C.) and U01DE024440 (R.A.S., O.D.K., B.H.). The UK Medical Research Council and Wellcome (Grant ref: 217065/Z/19/Z) and the University of Bristol provide core support for ALSPAC. A comprehensive list of grants funding is available on the ALSPAC website (http://www.bristol.ac.uk/alspac/external/documents/grant-acknowledgements.pdf). Funding for the collection of 3D face shape scans was specifically provided by the MRC and Wellcome Trust (092731) and the University of Cardiff. This publication is the work of the authors, and they will serve as guarantors for the contents of this paper. The funders had no role in study design, data collection and analysis, decision to publish, or preparation of the manuscript.

**Competing interests:** The authors have declared that no competing interests exist.

[14,16,19] and expanded sparse landmarks into spatially dense quasi-landmark representations of the face [4,5,7,8,20]. Then, starting from complete landmark configurations (sparse or dense), a popular feature extraction or phenotyping method is principal component analysis (PCA) to extract a set of orthogonal features that represent facial variation. More recently, alternative deep-learning networks, such as auto-encoders (AE), have emerged as non-linear counterparts to PCA. Despite the current trend favoring neural networks, to the best of our knowledge, these have not yet been applied in facial GWAS.

Apart from methods involving facial anthropometrics or unsupervised learning, supervised approaches have also been used to extract specific univariate facial features. For instance, it is feasible to extract facial characteristics expected to exhibit high heritability, such as facial traits shared among siblings [21]. Another illustration is GWASs conducted using resemblance scores guided by patient facial archetype associated with Achondroplasia [22] or Pierre Robin Sequence [23]. Similarly, resemblance scores to the distinctive facial endophenotype in unaffected relatives of individuals with non-syndromic cleft lip was successfully used in GWAS, which helped to further elucidate the genetic susceptibility to non-syndromic cleft lip [24].

Here, we provide a comprehensive comparison of univariate facial phenotyping approaches in GWAS of facial shape based on a cohort of 8,246 healthy European individuals. We evaluated phenotyping approaches based on two criteria: (1) GWAS discovery rate, defined as the number of independent association signals identified in aggregate across phenotypes in the same category (e.g., all principal components), and (2) SNP-based heritability determined by LDSC [9]. Additionally, this work offers secondary contributions by (1) exploring the latent dimensions of an AE as facial traits in GWAS, and by (2) introducing two additional supervised phenotyping schemes, one by extreme facial gestalts and another by randomly selected facial gestalts.

## Results

As illustrated in Fig 1, this study explored three distinct facial phenotyping strategies or categories. The first category, known as anthropometric techniques, focused on inter-landmark measurements. These measurements were defined as the Euclidean distances in 3D space between pairs of sparse facial landmarks. The second category, referred to as unsupervised techniques, involved deriving latent representations obtained through PCA and AE. These techniques generated up to 200 latent dimensions from spatially dense configurations of quasi-landmarks (n = 7,610), as established using MeshMonk [25]. The third category, termed supervised techniques, centered around resemblance-based facial traits, comparing each individual in the cohort to specific facial gestalts ranging from random to extreme to syndrome-related facial examples. Each face in the cohort received a resemblance score by measuring its cosine distance in multivariate face space against the provided facial examples (random, extreme, and syndromic). Generally, low phenotypic correlations were observed among different groups of phenotypes (Fig A in S1 File). All phenotyping methods were applied to the complete facial shape and, separately, to nasal shape. The focus on nasal shape was due to its high heritability, making it a particularly noteworthy facial region for detailed examination [26].

### SNP-based heritability

Fig 2 illustrates the distribution of SNP-based heritability, computed using LDSC [9], for facial traits extracted by various phenotyping methods (detailed descriptive statistics are provided in Table A in S2 File). For full facial shape, inter-landmark distances demonstrated the highest mean heritability, followed closely, without significant difference (Fig B in S1 File), by traits extracted through an AE. PCs and resemblance scores to randomly selected facial gestalts were

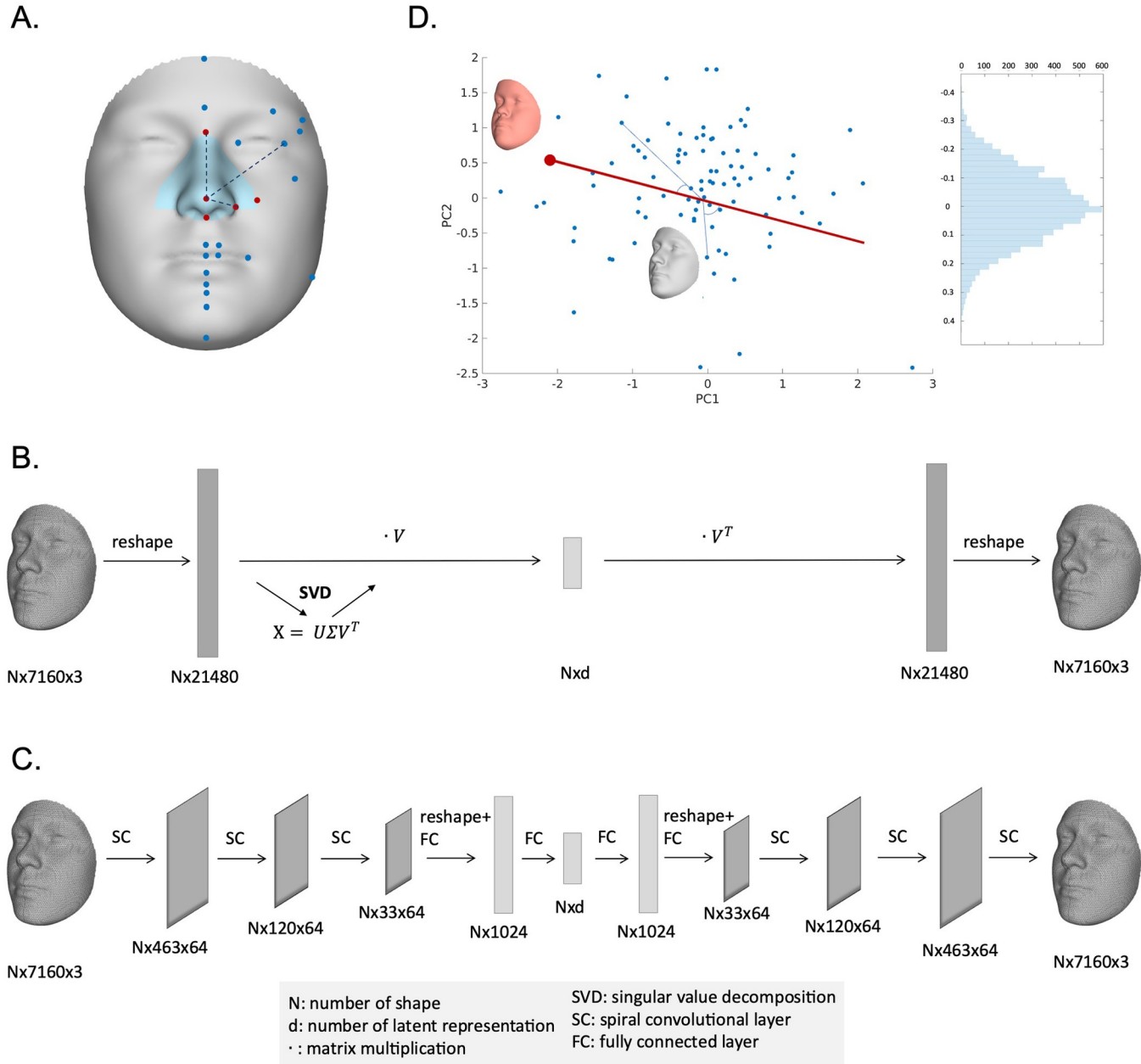

**Fig 1. Overview of phenotyping methods.** (a) inter-landmark Euclidean distances computed between 24 anatomical facial landmarks. The 5 nasal landmarks in the blue nasal region are highlighted in red. (b) principal component analysis, which is based on a low-rank singular value decomposition (SVD) applied to a reshaped representation of the 3D shape data, where matrix multiplication is denoted by ·. (c) an auto-encoder network. The encoder consists of three spiral convolutional layers, followed by two fully connected layers. The decoder architecture mirrors the structure of the encoder. (d) resemblance-based measures, defined as the cosine distance operating on the angle between the target vector (e.g., a random face, an extreme face) and a sample vector. For instance, a high resemblance to the averaged face of the achondroplasia cohort is demonstrated in red.

both ranked as the second most heritable traits, although PCs displayed greater variation in heritability scores. Notably, the mean heritability for resemblance scores to both extreme and syndromic facial examples was the lowest, implying a reduced influence of common genetic variants. Similar trends were observed for nasal shape, except that inter-landmark distances, in

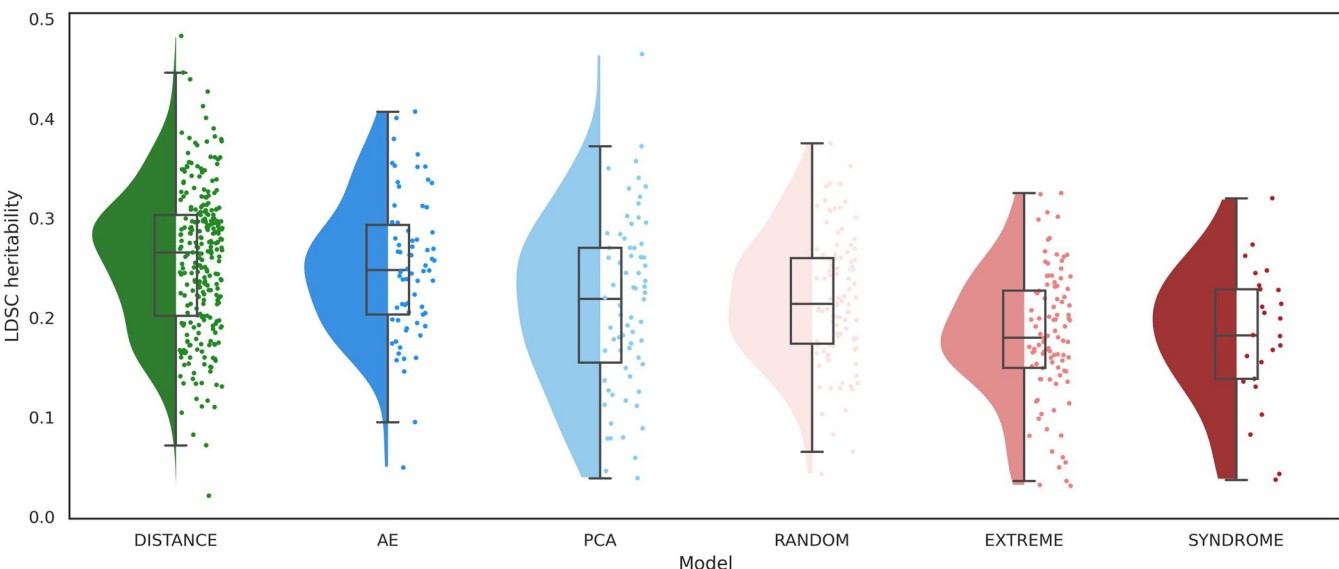

**Fig 2. Comparison of SNP-based heritability between phenotyping categories.** The colors represent different categories of traits: green for inter-landmark distances (DISTANCE), dark blue for traits extracted by auto-encoder (AE), light blue for traits extracted by principal component analysis (PCA), light red for resemblance scores to randomly selected facial examples (RANDOM), medium red for resemblance scores to extreme facial examples (EXTREME), and dark red for resemblance scores to syndrome facial archetypes (SYNDROME).

this scenario, displayed significantly higher heritability than all other categories of nasal phenotypes (Fig C in S1 File).

## Identification of trait-associated genetic loci

We assessed the GWAS discovery rate for various categories of facial traits by counting the number of independent genetic loci associated with a set of traits of the same type. We gradually increased the numbers of traits submitted for GWAS in each phenotype category, for example, the first N PCs, with N varying between 1 and the total number of PCs. Combining multiple univariate GWASs was achieved by taking the lowest P-value for each SNP across all the univariate traits considered. Furthermore, for each aggregation, we controlled for the additional multiple testing burden by estimating the number of independent traits (i.e., the effective number of traits) within the group. This adjustment allowed us to correct the genome-wide significance threshold (P < 5e-8) to a group-wide significance threshold as P < 5e-8 divided by the effective number of traits (Methods).

The effective number of traits within a single group is shown in Fig 3A. As expected, PCs are uncorrelated, so the number of independent traits equals the number of PCs used in a group. In contrast, inter-landmark distances exhibited a high degree of correlation, shown as a flattened curve. A lower degree of correlation was observed for resemblance-based traits (random/extreme/syndromic) and AE latent dimensions.

For each category of traits, the discovery rate generally increased when including more independent traits in GWAS (Fig 3B). This is most strongly observed for inter-landmark distances. For nasal shape, the limited number of 10 inter-landmark distances resulted in the poorest discovery rate overall. In contrast, 276 inter-landmark distances were extracted from full facial shape, leading to the best discovery rate across all tested measures.

For nasal shape, the findings for the unsupervised techniques of PCA and AE exhibited similar trends. Specifically, as more independent traits were included, the number of identified

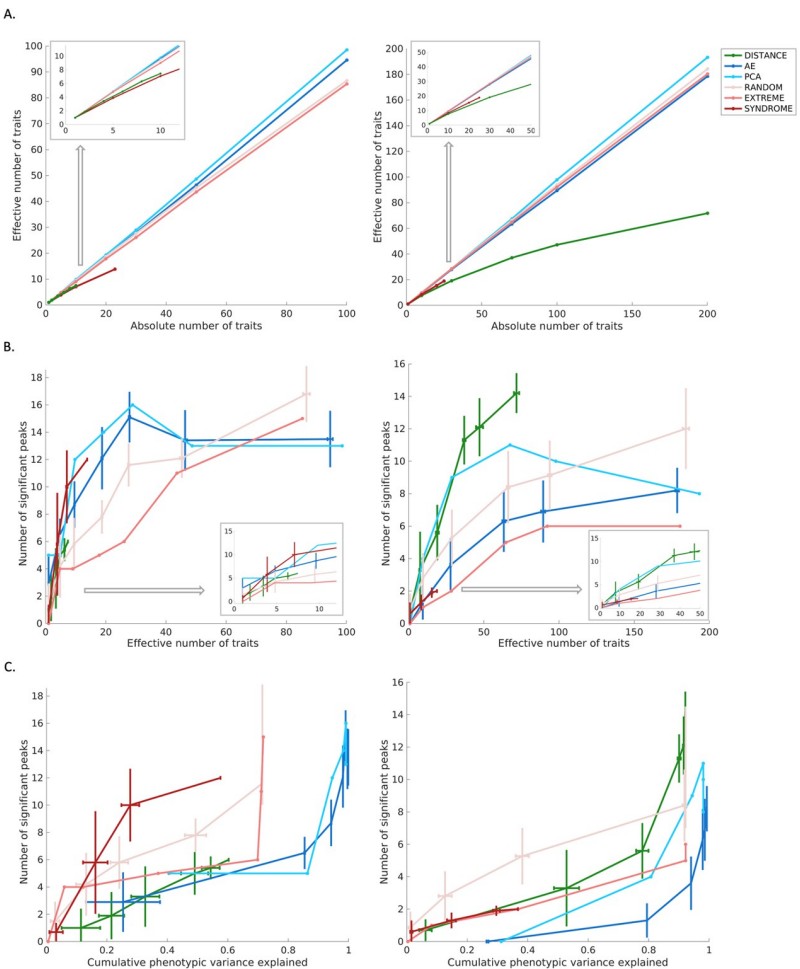

**Fig 3. The interplay among the dimensionality of traits, the number of significant genetic loci, and the phenotypic variation.** We compared nasal shape phenotypes (left columns) and full facial shape phenotypes (right columns) in terms of (a) the effective number of traits, (b) the effectiveness of identifying independent genetic loci through GWAS, and (c) the phenotypic variation captured by traits and their corresponding number of significant genetic loci in GWAS. For nasal shape, the experiments were conducted with absolute numbers of traits equal to [1, 5, 10, 20, 30, 50, 100]. Since there were a limited number of inter-landmark distances and syndromic groups, the absolute numbers of traits were set to [1, 2, 4, 6, 8, 10] and [1, 5, 10, 23], respectively. Similarly, for facial shape, the experiments were conducted with absolute numbers of traits equal to [1, 10, 30, 70, 100, 200]. The absolute numbers of traits based on resemblance to syndrome gestalts were set to [1, 10, 20, 25]. The colors represent different categories of traits: green for inter-landmark distances (DISTANCE), dark blue for traits extracted by auto-encoder (AE), light blue for traits extracted by principal component analysis (PCA), light red for resemblance scores to randomly selected examples (RANDOM), medium red for resemblance to extreme examples (EXTREME), and dark red for resemblance scores to syndromic examples (SYNDROME). Unlike PCs, which are ordered according to descending explained variance, and resemblance scores to extreme gestalts based on the cosine distance to the mean shape, there is no specific order within other categories of traits. Therefore, given a fixed absolute number of traits, we randomly selected a subset 10 times from the full set of inter-landmark distances and resemblance to syndromic gestalts. Additionally, 10 replicates were performed for generating multiple AE latent dimensions and resemblance to random gestalts under different random initializations. The error bars represent the variation in results obtained from these 10 replicates.

genetic loci initially increased until it reached a maximum, after which a decline in the discovery rate was observed. This decline can be attributed to the tradeoff between adding less genetically interesting traits and a more significant threshold that is required to adjust for multiple testing. Particularly in the case of PCA, it is well-established that later PCs primarily model noise in the data and are not expected to contribute to further genetic discoveries. The same

was observed for the latent dimensions of AE, despite their lack of a specific order in terms of phenotypic variance explained, unlike PCs. For full facial shape, a similar pattern of initial increase and subsequent decline was observed for AE and PCs, but the AE latent dimensions failed to reach the same discovery rate as PCs.

For the supervised techniques, the relatively small number of syndromes (n = 25) may have impacted the overall GWAS discovery rate for this group when compared to all the other phenotyping strategies. Nonetheless, in the case of nasal shape, the maximum discovery rate for syndrome archetypes is high compared to the number of independent traits used. Conversely, this was not the case for full facial shape. This finding highlights that syndrome archetypes are valuable, particularly in nasal regions, but may not be as effective in characterizing full facial variation. The outcomes obtained by extreme facial gestalts initially showed a lower identification rate of associated common variants, but gradually converged with other techniques as the number of independent traits increased. It is important to note that this convergence is essentially a result of treating more faces as "extreme", even though they may actually be less or no longer extreme (as explained in the Methods). Lastly, in the case of resemblance to random facial gestalts, a steady increase in GWAS discovery rate is observed as the number of independent traits increases. Notably, when further expanding the number of random facial gestalts used (Fig D in S1 File), this approach outperforms all other methods. In other words, the benefits of adding more traits outweigh the multiple testing burden in this scenario. However, due to the randomness involved, the GWAS discovery rate showed greater variation when repeating the experiment over consecutive runs, as indicated by the error bars in Figs 3B and 3D in S1 File.

Fig 3C illustrates the GWAS discovery rate plotted against the cumulative phenotypic variance explained by each phenotyping method. The variance explained for a group of facial traits was measured using partial least-squares (PLS) regression (using the 'plsregress' function from MATLAB R2022b) with the original images (3D quasi-landmark configurations) as responses and the grouped univariate facial traits as predictors. The cumulative variance of all PLS components reflects the explained phenotypic variance. Interestingly, the first PC, while explaining 31.22% of the phenotypic facial variation, did not yield any significant genetic loci. Furthermore, the first 10 PCs captured 80.75% of total facial variation but resulted in the identification of only 4 independent genetic loci. The same was observed for AE dimensions. This suggests that, while a substantial amount of geometric phenotypic variance is captured by the first few PCs and AE dimensions, they do not necessarily correspond to genetically relevant information. In contrast to both dimensionality reduction techniques, the number of identified genetic loci based on inter-landmark distances and resemblance-based scores increased rapidly with even a limited number of traits, explaining only a few percent of the complete facial variation. This indicates that, while these traits capture less geometric facial variation, they result in a greater number of discoveries in GWAS, suggesting that these traits are enriched for genetically determined aspects of shape variation.

## Sharing of genomic signals

We tested whether various types of traits resulted in overlapping or distinct sets of identified independent genetic loci and annotated genes (Fig 4 and Tables B-E in S2 File). For each group of traits, we evaluated genetic loci under the "best-case scenario", i.e., when the maximal number of independent genetic loci was reached. Genetic loci were considered shared between two different methods if their respective lead SNPs were located within 250kb of each other. The choice of a 250kb window originates from the default settings in FUMA (SNP2GENE) for determining independent loci. Considering that AE latent dimensions and randomly selected

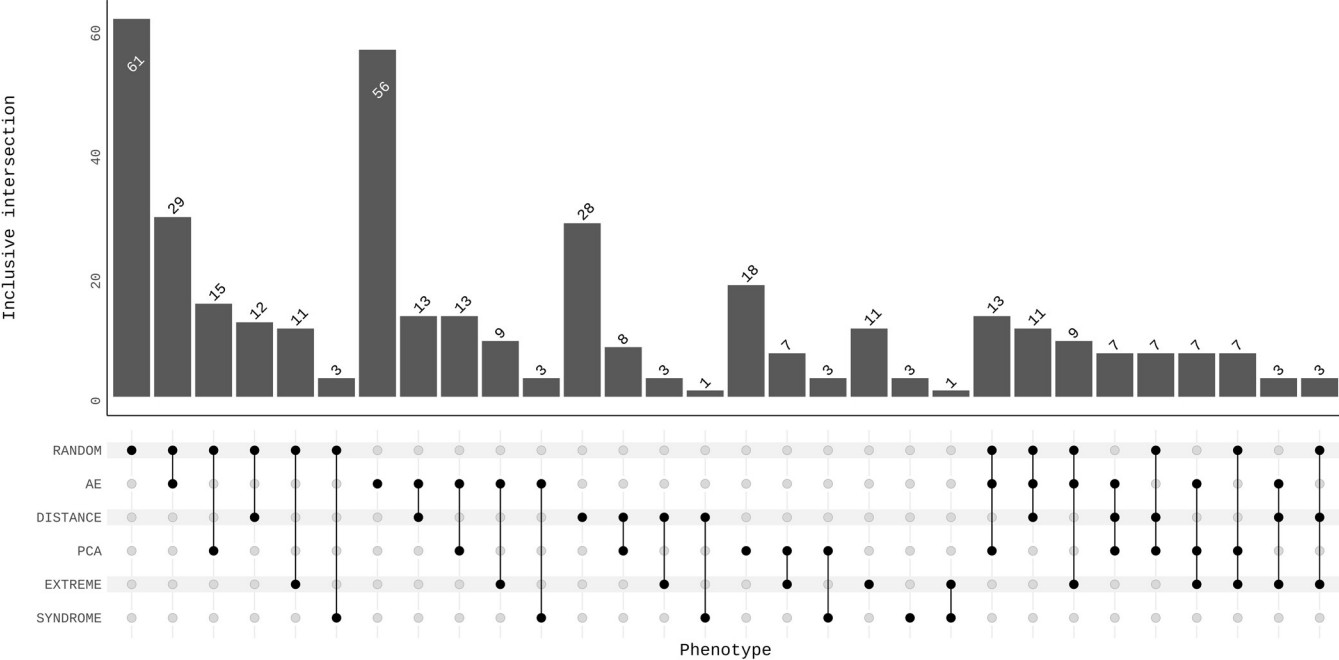

**Fig 4. Comparing phenotypes in terms of overlapping genetic findings from facial GWAS.** The number of overlapping genes annotated using GREAT is displayed. The significant genetic loci were identified using the optimal number of independent traits, i.e., when the number of independently significant genetic loci after multiple testing correction was at its maximum. Phenotypes include inter-landmark distances (DISTANCE), traits extracted by auto-encoder (AE), traits extracted by principal component analysis (PCA), resemblance scores to randomly selected examples (RANDOM), resemblance scores to extreme examples (EXTREME), and resemblance scores to syndromic examples (SYNDROME). To account for the variability introduced by random initializations in AE and resemblance scores to randomly selected gestalts, we used the union set from 10 replicates to intersect with other trait groups. The patterns are first grouped by phenotype categories and then sorted by frequency, starting with all pairwise overlaps (21 combinations across 6 groups) and extending to the top 9 three-way overlaps, covering a total of 30 sharing patterns. Additionally, all possible overlap combinations, including up to six-way overlaps, are detailed in Table B-E in S2 File for further reference.

facial gestalts are inherently stochastic phenotyping strategies, we conducted multiple runs for these approaches to assess the impact of randomness on the results. Specifically, we examined the degree of overlap and overall consistency of the identified loci across 10 replicates for both AE and resemblance to random gestalts (Tables F-I in S2 File). The union set of genetic loci from these 10 replicates were then used to intersect with other trait groups.

Surprisingly, the extent of overlap in terms of genetic loci between different methods was relatively limited (Tables B-C in S2 File; LocusZoom plots available in the source data). For facial traits, the highest pairwise intersection over union (IoU) was 0.33, with a median of 0.17. For nasal traits, the highest pairwise IoU was 0.43, with a median of 0.22. When taking the union of all independent genetic loci identified across different approaches, we found 60 loci associated with the nose and 58 loci associated with the face. This suggests that each of the phenotyping strategies capture distinct aspects of facial shape variation and, as a result, they strongly complement each other in pinpointing genetic factors that influence facial shape.

Similarly, for 10 replicates of generating AE latent dimensions and resemblance to random gestalts based on full facial shape, the union set of identified genetic loci across all 10 randomizations yielded 31 and 33 genetic loci, respectively. The highest pairwise IoU for AE latent dimensions across 10 runs was 0.56, with a median of 0.31. There was no significant difference in the mean IoU between the intersections of PCA and 10 AE runs, and the intersections within multiple AE runs ($p = 0.9$ from a two-sample t-test), suggesting that the inconsistency of multiple AE runs is comparable to the variation between different phenotyping methods. A

similar conclusion holds for resemblance to random gestalts, where the highest IoU reached 0.58, with a median of 0.31. Detailed information on overlapping genetic loci across all combinations of the 10 replicates for nasal and facial shape is provided in Tables F-I in S2 File.

The number of pairwise overlapping genes followed a similar pattern to the number of pairwise overlapping genetic loci, as expected (Fig 4 and Tables D-E in S2 File). Several genes encoding key craniofacial transcription factors, including *ALX1*, *PAX3*, *TBX15*, and *SOX9*, were consistently identified, regardless of the category of traits used (Fig E in S1 File). The complete list of genes detected by at least four different categories of traits (out of the total of six groups) can be found in Table J-K in S2 File. When considering a single trait, the identification of genes was relatively constrained, resulting in a corresponding limitation in detecting Gene Ontology (GO) biological processes. However, based on the union set of lead SNPs from all groups of phenotypes, the top terms (based on lowest binomial P values) in the GO biological processes category were all highly relevant to craniofacial shape (full lists can be found in source data). This again indicates the idea that different phenotyping strategies are indeed complementary in capturing the diverse genetic influences on craniofacial shape.

## Discussion

In this study, we evaluated and compared different techniques for extracting univariate facial phenotypes in humans, quantified from 3D facial images. Traditional anthropometric traits, such as inter-landmark distances, demonstrated the highest mean heritability suggesting that they are well focused towards genetically determined aspects of shape variation. While the set of inter-landmark distances yielded a relatively high number of GWAS loci compared to a similarly sized set of traits from a different phenotyping category, the total number of loci identified was ultimately limited by the number of available landmarks. This became especially apparent for nasal shape, where only 5 landmarks were available to extract pairwise distances, such that all other phenotyping categories identified a greater number of GWAS loci. Even though the absolute number of inter-landmark distances rapidly increases with each additional landmark, the number of independent phenotypes lags behind due to the high degree of correlation between these measurements. Therefore, the scalability of this phenotyping approach is limited at a computational cost. This may partly be alleviated by selecting the most accurate and distinctive measures based on prior knowledge of anatomy and biology [27,28]. Altogether, measuring inter-landmark distances, already used extensively in facial genetics [1,2,10–18,29], is a viable univariate phenotyping method with a good yield in GWAS on the condition that enough landmarks are available and computational cost is considered. However, in comparison to the other techniques, they are highly correlated and are likely to identify only a specific set of genetic influences to facial shape. Therefore, it is ideal for this approach to be supplemented with another strategy to cover the full spectrum of genetic factors underlying facial shape.

A more complete description of facial shape can be obtained by modeling the set of dense 3D quasi-landmark coordinates, which constitutes a highly correlated set of facial features. Unsupervised dimension reduction techniques offer a means to compress this set into a reduced set of morphological variables that can be used as traits in GWAS analysis thereby using dramatically fewer computational resources compared to using the individual landmarks.

Among the unsupervised dimension reduction methods for facial shape analysis, PCA has seen the most use in the literature, including in GWAS analysis [3]. PCA is deterministic, conceptually simple, and available in most data analysis platforms. One advantage that PCA offers is the ordering of its PCs according to their contribution to phenotypic variance. It is well-

established that noise from the original images is modeled by the later PCs, which makes it straightforward to determine how many PCs to retain post hoc. However, we observed that the amount of phenotypic variance explained by a single PC does not necessarily indicate its utility for discovering genetic associations. For example, a GWAS on the first PC of facial shape failed to identify a single locus, despite this PC explaining 31.22% of overall shape variation. In fact, when looking at the combined GWAS results across all the facial shape PCs (Fig 3B and 3C), we observed that the majority of independently identified genomic loci were contributed by PCs 10–40. Earlier PCs explained more phenotypic variation but did not identify as many genetic associations. Later PCs (>40) did not contribute many additional loci but did exacerbate the multiple testing burden, resulting in an optimal number of loci identified at around 70 facial PCs followed by a drop-off. Furthermore, we found that PCs exhibit a lower mean heritability compared to inter-landmark distances with a wide range in heritability values across the PCs. This may suggest that, while some components have a strong genetic basis, others may not. This may be attributed to by the fact that PCs are essentially mathematical constructs constrained to be mutually orthogonal, whereas inter-landmark distances have the freedom to be correlated, capturing slightly different yet overlapping information. Altogether, PCs derived from dense landmark configurations almost fully capture the available 3D shape information and are straightforward to acquire. However, we have shown that the order of features/PCs based on phenotypic variance explained does not necessarily indicate their relevance for genetic findings.

Another dimension reduction technique considered in our study was an AE. These deep learning-based networks have surfaced as a popular non-linear alternative to PCA in many fields of research including image analysis [30,31]. However, the latent dimensions of an AE are currently underexplored as a phenotyping strategy, and have never, as far as we are aware at the time of writing, been used in facial GWAS analysis. In contrast to PCA, setting up and training an AE network requires far more time and expertise due to its complexity and the extensive parameter tuning required. For example, the number of latent variables needs to be set prior to model training, and creating more compact or elaborate models requires re-training. Simply excluding latent dimensions leads to poor reconstruction performance [32], hence determining the optimal latent dimensionality becomes a process of trial and error. Furthermore, latent variables of an AE are unordered, explain similar amounts of overall phenotypic variation, can encode for non-linear data interactions, and are not subject to any orthogonality constraints. These properties have likely contributed to their high SNP-based heritability, only second to inter-landmark distances and significantly higher than PCs. However, despite their high expected SNP-based heritability, AE latent dimensions identified a similar number of independent genomic loci in GWAS on nasal shape compared to PCs, and fewer in GWAS on facial shape. These results suggest that although individual AE dimensions may have a strong genetic basis, properties such as their cross-correlations and redundancy make them no better than PCs for genetic discovery. These observations challenge the increasing preference for machine learning-based algorithms in facial analysis, where PCA is criticized for relying on linear transformations and therefore likely struggling with non-linearity in facial data. However, non-linearity might not be as abundant as one might expect in static facial shapes (Fig F in S1 File), or alternatively, the added value of this ability is only minimal in the context of GWAS. This is unlike situations where machine learning algorithms have outperformed PCA by learning the nonlinear variations associated with different facial expressions or pose conditions [33].

While dimension reduction methods are powerful for extracting features from high-dimensional correlated datasets, the biological meaning of their resulting features and the validity of the results reported in the field of genetics have been questioned [34,35]. To ensure biological relevancy of the obtained morphological variables, some studies [19,36] have first derived

phenotypes through a dimension reduction method and subsequently selected a subset of traits for downstream analysis based on heritability estimations. A more sophisticated approach adopted by some recent studies is to rely on prior biological knowledge to derive likely heritable facial traits in a supervised manner. Focusing on heritability directly, researchers have extracted highly heritable facial traits by considering familial resemblance and family-based heritability estimations from which they derived measures such as the principal component of heritability [37–39] and siblings-shared facial traits [21]. Furthermore, to investigate both typical-range and disease-associated variation in facial morphology, some studies employed a phenotyping method supervised by genetic conditions characterized by distinct facial features. Examples include resemblance scores to the facial archetype associated with Achondroplasia [22] and Pierre Robin Sequence [23]. This approach directly measures the facial features that result from subtle variations within the same physiological pathways, which when disrupted result in distinct (sub-) clinical facial characteristics. Substantially expanding on this approach, our comparative study included resemblance scores supervised by the facial archetypes derived from 25 syndromes associated with distinct facial characteristics.

In addition to syndrome-driven phenotypes, extreme phenotypes—defined as dichotomized scores by comparing individuals with relatively extreme PC scores to those without—were initially explored by Crouch et al. [19] and found to be associated with large-effect single gene variants. Building on this insight, we recognized that multidimensional facial variations allow for the identification of extreme (but non-clinical) faces. These extreme faces can also be used to supervise resemblance scores, now using a continuous measure to evaluate the presence of a specific extreme facial pattern in an individual. Additionally, a randomly selected actual face is expected to reflect genetic signals, as it is a product of inheritance. Therefore, we further generalized this approach to supervise facial phenotyping using randomly selected facial examples.

Resemblance scores supervised by syndromic facial archetypes exhibited lower mean heritability and resulted in fewer genetic loci compared to other groups of traits. This may be explained by the limited number of syndrome groups and the role of low frequency genetic variants. To illustrate, the limited number of syndrome groups resulted in a limited number of syndrome-derived traits, further leading to a lower statistical power. In addition, as GWASs focus on common genetic variants, they overlook low-frequency and rare genetic variants that could potentially underpin these traits. Similar findings were observed for resemblance scores to extreme facial gestalts. While eventually achieving a comparable GWAS discovery rate to PCA, this convergence primarily resulted from the inclusion of more extreme facial examples, which were progressively less extreme. Nevertheless, while resemblance scores derived from syndromic and extreme facial examples may not yield the greatest number of loci in GWAS, studies [22–24] have demonstrated that a targeted facial phenotyping resulted in GWAS loci that displayed a stronger link with disease etiology versus non-targeted phenotyping approaches. Therefore, facial traits derived from genetic conditions may facilitate the discovery of disease-related genes and pathways in future investigations. This could be especially interesting in the context of uncommon and rare genetic variants available from whole-exome or whole-genome datasets.

Resemblance scores to random facial gestalts surpassed all the other phenotyping approaches in terms of the number of identified genetic loci in GWAS, on the condition that enough of such traits were considered. Measuring the resemblance to a specific randomly selected facial gestalt can be thought of as measuring the extent to which a specific person's set of facial features is present in the faces of the other individuals within the cohort. Therefore, the total number of extractable traits is equal to the cohort size, usually in the thousands. Mathematically, each randomly selected facial gestalt, under the absence of identical twins,

represents a unique direction in the face space, thus allowing one to sample that space in a brute-force-like way. Compared to other phenotyping approaches, these traits displayed a high mean SNP-based heritability and yielded a high number of significant genetic loci relative to their explained phenotypic variance. Together, this suggests that a measure of resemblance to a random facial gestalt captures genetically determined aspects of facial shape variation. A possible explanation could be that this approach intentionally focusses on facial features that are observed within a cohort as a result of inheritance, rather than on purely mathematical decompositions of facial shape. In summary, the ability to generate many facial phenotypes with a high expected heritability and that yield a set of complementary loci in GWAS, make resemblance to randomly selected facial gestalts a great option for those willing to accept the computational burden.

The limited overlap observed in identified genetic loci across different methods suggests that each phenotyping strategy captures distinct genetic factors influencing facial shape, which is consistent with the generally low phenotypic correlations (Fig A in S1 File). This observation may reflect the Beavis effect [40, 41], where each method samples from a larger, underlying but truncated distribution of biologically real signals, and the detected loci are subsample specific. The more underpowered a study is to capture the full range of effects, the more pronounced the Beavis effect becomes, increasing the probability of non-replication of genuine signals. In other words, with unlimited and continuously growing sample sizes, it might become possible that the different phenotyping strategies converge onto each other, and that genetic loci identified by one strategy are replicated by another strategy. However, with the current sample sizes of today, that remains to be investigated.

When using resemblance scores for random gestalts and AE latent scores, the sets of identified genetic loci varied substantially across multiple replicates of GWAS due to different random initializations (Tables F-I in S2 File). This highlights the importance of conducting multiple runs, as the inherent randomness in the process proves advantageous in thoroughly exploring the entire spectrum of facial shape variation. Although this introduces challenges for interpretation and replication, the larger union of significant loci provides valuable opportunities for a more comprehensive investigation into the genetic basis of facial shape variation. These observations also suggest the possibility of optimization. For example, it could be valuable for future studies to investigate how to generate a minimal set of facial traits that maximizes genetic findings thereby alleviating some of the computational burden. Nonetheless, regardless of the category of phenotypes used, key craniofacial transcription factors were consistently identified, and the combined set of loci across all phenotyping categories yielded GO biological processes that were highly relevant to craniofacial shape. This underscores that different phenotyping approaches complement each other in the identification of genetic factors influencing facial shape.

In this comprehensive study, we conducted a thorough evaluation of various univariate phenotyping methods for the characterization of human facial shape. These methods were categorized into three groups, which encompassed anthropometric traits, traits derived through unsupervised dimension reduction techniques, and supervised resemblance-based traits. Our findings expand the current understanding of the genetic relevance of various univariate traits, including their SNP-based heritability and GWAS discovery rates. Traditional anthropometric traits, which are derived from a set of landmarks with clear anatomical meaning, exhibit high SNP-based heritability, making them suitable traits for genetic investigations. Though, their limitation mainly lies in their fundamentally incomplete morphological description, especially when the number of landmarks is limited. On the other hand, dimension reduction methods, which despite lacking a clear biological meaning, can more fully capture morphological variation and subsequently identify a good number of genomic loci in GWAS. However, our

analyses have shown that for the purpose of GWAS analysis, training an AE network is likely not worth the hefty time investment as it identified fewer independent genomic loci compared to PCA. As an alternative, our study has expanded on the idea of supervised resemblance-based phenotypes by using facial gestalts from 25 genetic conditions as well as randomly selected and extreme, non-clinical facial gestalts. While resemblance scores to randomly selected facial gestalts are easy to acquire and have demonstrated their potential to capture genetically relevant facial shape variations in GWAS, resemblance scores to extreme and syndromic facial gestalts may be useful in the search of rare genetic variants in future studies. Overall, this work investigated various types of univariate phenotyping strategies for facial shape, which could potentially be extended to other morphological structures, such as brain shape, providing valuable references for future research.

## Materials and methods

### Ethics statement

We have complied with all relevant ethical regulations for work with human participants and informed consent was obtained. Institutional review board (IRB) approval was obtained at each recruitment site and all participants gave their written informed consent prior to participation; for children, written consent was obtained from a parent or legal guardian. For the 3DFN sample, the following local ethics approvals were obtained: Pittsburgh, PA (PITT IRB PRO09060553 and RB0405013); Seattle, WA (Seattle Children's IRB 12107); Houston, TX (UT Health Committee for the Protection of Human Subjects HSC-DB-09-0508); and Iowa City, IA (University of Iowa Human Subjects Office IRB (200912764 and 200710721). For the Penn State sample, the following local ethics approvals were obtained: Urbana-Champaign, IL (PSU IRB 13103); New York, NY (PSU IRB 45727); Cincinnati, OH (UC IRB 2015–3073); Twinsburg, OH (PSU IRB 2503); State College, PA (PSU IRB 44929 and 4320); Austin, TX (PSU IRB 44929); and San Antonio, TX (PSU IRB 1278). For the IUI sample, the following local ethics approvals were obtained: Indianapolis, IN and Twinsburg, OH (IUI IRB 1409306349). For the ALSPAC sample, ethical approval for the study (Project B2261: "Exploring distinctive facial features and their association with known candidate variants") was obtained from the ALSPAC Ethics and Law Committee and the Local Research Ethics Committees. Consent for biological samples was collected in accordance with the Human Tissue Act (2004). For the syndromic face dataset, this study was approved by the ethical review board of KU Leuven and University Hospital Leuven (S60568, Leuven, Belgium).

### Dataset and preprocessing

The analysis included participants with typical-range facial shape of European descent from independent population-based cohort studies conducted in the United States (US, $n_{US}$ = 4,680) and the United Kingdom (UK, $n_{UK}$ = 3,566). In our previous work [5], this dataset (referred to as the EURO dataset) was used for a multivariate GWAS study on facial morphology. The US samples originated from three independent data collections: the 3D Facial Norms cohort [42] (3DFN) and from studies at the Pennsylvania State University (PSU) and Indiana University Indianapolis (IUI). The UK samples were part of the Avon Longitudinal Study of Parents and their Children [43,44] (ALSPAC). Information on the different genotyping platforms, imputation, and quality control can be found in [5]. Intersection of imputed and quality-controlled SNPs across the US and UK datasets yielded 7,417,619 SNPs for analysis. The 3D facial surface images were registered using the MeshMonk [25] registration framework in MATLAB (R2017b) as described in [5]. In total, 8,246 unrelated participants with recent

European ancestry passed genotyping, imaging, and covariate quality control, and were used for analysis.

We used a subset from the syndromic face dataset in our previous work [45], where it was originally applied for a syndrome classification task. This subset was obtained from two databases: 1) the FaceBase repository "Developing 3D Craniofacial Morphometry Data and Tools to Transform Dysmorphology, FB00000861" [46]; 2) Peter Hammond's legacy 3D dysmorphology dataset hosted at the KU Leuven, Belgium [47]. Syndromes can be categorized based on whether the underlying genetic conditions can be diagnosed based on typical facial characteristics [45]. In this study, we focused on syndromes with typical facial features falling into category A and B as defined in [45], including 25 out of the total 51 syndromes (Table L in S2 File). Overall, there were 1,784 3D syndromic facial images and a control group of 54 individuals unrelated to patients with known genetic syndromes. These control images were used to determine whether the average syndromic images were significantly different from those of the healthy controls for each syndrome group.

The 3D facial surface meshes, comprising 7,160 dense quasi-landmarks were aligned using generalized procrustes analysis (GPA), symmetrized, and subsequently adjusted for age, age-squared, sex, weight, height, facial size, camera system, and the first 4 genomic ancestry PCs using PLS regression (function 'plsregress' from MATLAB R2022b). The same procedure was performed independently for the nose, which was obtained by applying the data-driven hierarchical facial segmentation method described in [4,5]. Essentially, facial segments were defined by grouping strongly correlated vertices using hierarchical spectral clustering [4,48]. The strength of correlation between quasi-landmarks was measured using Escoufier's RV coefficient [49,50]. Subsequently, the RV coefficient was used to construct a similarity matrix that defined the formation of facial segments. As shown in Fig 1A, the highlighted nose module consists of 758 vertices.

## Facial phenotyping strategies

In this study, we explored three categories of phenotyping methods: the first category involved anthropometrics traits, exemplified by inter-landmark distances; the second category encompassed latent scores derived through dimensionality reduction methods such as PCA and AE; and finally, resemblance-based traits were defined as the 1 - cosine of the Mahalanobis angle between the vectors of the target sample (extreme/syndromic/random gestalts) and each sample in the EURO cohort.

## Inter-landmark distances

Since the images were symmetrized, we focused on 24 anatomical facial landmarks on the right half of the face, including the facial midline (Fig 1A). Most landmarks have been used in previous GWASs of facial variation and have shown relatively high heritability [10,17]. The phenotypes were computed as inter-landmark Euclidean distances between landmarks (in total 276 for face, 10 for nose). We followed a semi-automatic landmarking procedure as described in [25] using MeshMonk to position the landmarks onto all samples. First, a set of randomly selected facial scans (N = 5) was manually landmarked three times by two observers. Subsequently, the average positions among iterations were calculated for each landmark, and the resulting placements were transferred to the template through barycentric coordinate conversion. These average placements on the template served as the foundation for the automated landmark placements. Finally, since the faces are in the same coordinate system as the original template, the averaged landmark positions could be automatically transferred to the entire

dataset. The facial template in Wavefront (.obj) format, the coordinates of 24 facial landmarks and 5 nasal landmarks on this template can be found in source data.

**Unsupervised dimensionality reduction of dense quasi landmarks.** *Principal component analysis*. Principal component analysis (PCA) simplifies complex facial variation by transforming high-dimensional mesh configurations into a small number of uncorrelated features, i.e., principal components (PCs). The original dense landmark configurations were structured into a three-dimensional matrix with dimensions $N$ (number of shapes), $L$ (7,160 quasi-landmarks), and 3 (x-, y-, and z-coordinates of each landmark). To perform PCA, we first mean-centered the data and reshaped it into a two-dimensional matrix with dimensions $N \times 3L$. Subsequently, we applied low-rank singular value decomposition (SVD) to the mean-centered reshaped data matrix $X \in \mathbb{R}^{N \times 3L}$, defined as $X = U\Sigma V^T$ (Fig 1B). The diagonal matrix $\Sigma$ contained the singular values and the columns of U and V consisted of the left and right singular vectors, respectively. The right singular vectors in V represented the PCs. Additionally, PCA was performed in combination with parallel analysis [51, 52] to capture the major shape variance with the optimal number of variables. This resulted in 32 PCs explaining 99.21% of nasal shape variation and 70 PCs explaining 98.08% of facial shape variation.

*Auto-encoder*. An auto-encoder (AE) works as a non-linear generalization of PCA, comprising two main parts: an encoder and a decoder. The encoder compresses the data into a small number of variables and the decoder aims to reconstruct the original data from that compact representation. The advantage of using an AE is that it can model non-linear relationships that may be present in the data. However, as opposed to PCA, the disadvantage of an AE is that the latent variables are not necessarily uncorrelated.

Fig 1C shows the structure of the auto-encoder network used to extract features based on 3D facial meshes as previously used in [53]. The first several layers of the encoder consist of spiral convolutional layers, which reduce the size of the input. Each spiral convolutional layer consists of a spiral convolution operator and a mesh simplification step. Spiral convolution operators [54,55] are analogous to the grid-based convolutional filters in traditional convolutional neural networks and are designed as spirals starting at a center point and proceeding outwards from a random adjacent point. The mesh simplification step reduces the input size based on a predefined fixed scheme, achieved by performing quadric edge collapse on the template using MeshLab software [56]. The three spiral convolutional layers consist of 64, 64, and 64 learned filters, respectively, followed by the addition of two fully connected layers to further compress the data into the desired number of latent variables. The decoder architecture mirrors the encoder architecture. The model is trained to minimize the reconstruction error. Training strategy and implementation details can be found in S1 Methods.

## Supervised resemblance measurements

Individual faces can be represented as single points or vectors situated in a multidimensional "face space", where each dimension reflects a continuous axis of morphological variation [57,58]. To construct such a face space, we applied PCA to the symmetrized and GPA aligned quasi-landmarks of the 8,246 samples, as mentioned above, and retained an equal number of PCs for consistency, i.e., 32 PCs explaining 99.21% of nasal shape variation and 70 PCs explaining 98.08% of facial shape variation. Note that, in principle, a shape space can alternatively be obtained using a different dimension reduction method. In our space, each face could be represented as a vector encoding the scores along each PC. In other words, the vector representation of a single face represented the extent to which the facial features encoded by each PC were present within that face. Following the idea that the resemblance between two faces can be measured by the correlation between their features, we quantified the facial

resemblance of one face to another as the cosine distance derived from the angle enclosed by their feature vectors in a Mahalanobis standardized space (Fig 1D) [59]. To obtain resemblance-based scores for GWAS analysis, we calculated facial resemblance scores between each face from the cohort and a specific facial example, whereby we considered different possibilities for the choice of facial example.

In a first scenario, we considered the facial example to be a randomly selected face from the cohort and calculated resemblance-based facial phenotypes for GWAS as the cosine distance between the vector of the EURO cohort faces and the vector of the selected random facial example. We gathered additional resemblance-based facial phenotypes by selecting additional randomly selected facial examples. A second category includes the resemblance of the EURO cohort to an extreme facial example. To do so, we first ranked all the individuals based on their Mahalanobis distance from the estimated mean face, which could be represented as the origin of the face space. Subsequently, we selected the top k (desired dimension) individuals that were located most peripherally in the face space. Each sample from the EURO cohort was then scored by computing the cosine distance between its vector and the vector of each individual extreme facial example. A third category included resemblance to syndromic faces. We projected 1,784 syndromic faces from 25 distinct syndromes into the learned PCA space based on the EURO cohort and computed the average shape from each syndrome group. Using a permutation testing framework as described in [23], we tested which of the average syndromic faces were significantly different from the healthy controls and subsequently removed any syndromes without any distinct (P>0.05) characteristics (n = 0), leaving 25 for further analysis. We repeated this procedure for the nose, where 23 out of 25 syndromes were considered for further analysis (details of syndrome groups in Table L in S2 File). Resemblance-based phenotypes for GWAS were obtained by measuring the cosine distance between the EURO cohort and the syndromic facial gestalts, which were calculated as the average face per syndrome.

## Genome-wide association meta-analysis

For each univariate trait, GWASs were conducted in the US and UK cohorts independently using linear regression (function 'regstats' from MATLAB 2022b) where SNPs were coded under the additive genetic model (0, 1, 2). The SNPs were adjusted for covariates (sex, age, age-squared, height, weight, facial size, the first four genomic ancestry axes and the camera system), prior to the linear regression. This generated effect size and standard error estimates for the US and UK cohort separately which were then meta-analyzed using the inverse-variance weighted method [60]. Meta P-values were computed using a two-tailed test.

## Aggregation of multiple GWAS studies

To investigate the number of identified genetic loci under different numbers of traits, we gradually increased the absolute numbers of traits in each phenotype category. For nasal shape, the experiments were conducted with absolute numbers of traits equal to [1, 5, 10, 20, 30, 50, 100]. Since there were a limited number of inter-landmark distances and syndromic groups, the absolute numbers of traits were set to [1, 2, 4, 6, 8, 10] and [1, 5, 10, 23], respectively. Similarly, for facial shape, the experiments were conducted with absolute numbers of traits equal to [1, 10, 30, 70, 100, 200]. The absolute numbers of traits based on resemblance to syndrome gestalts were set to [1, 10, 20, 25].

To aggregate multiple GWASs on univariate traits within a phenotype group, we employed Tippett's minimal-p meta-approach [61]. Furthermore, for each aggregation, we controlled for the additional multiple testing burden by estimating the number of independent traits (i.e., the effective number of traits) within the group. This adjustment allowed us to correct the

genome-wide significance threshold (P < 5e-8) to a group-wide significance threshold (5e-8 divided by the effective number of traits). Since PCA yielded mutually uncorrelated univariate features, the number of independent phenotypes was equal to the number of PCs used. For all other methods, this number was estimated using permutation testing [62]. Specifically, each of 7,417,619 SNPs was randomly permuted and the same GWASs were repeated once. This allowed to estimate the null-distribution of the minimum P-values for each SNP across the set of univariate traits. The number of independent phenotypes was then estimated as 0.05 divided by the 5th percentile of this null distribution [62].

### SNP-based heritability estimation

SNP-heritability is defined as the proportion of phenotypic variance that is explained by additive genetic effects of SNPs. First, SNPs were intersected with the HapMap3 SNPs and any SNP with non-matching alleles was removed, as well as SNPs within the major histocompatibility complex region. The SNP heritability of each univariate trait was then estimated with LDSC (published software https://github.com/bulik/ldsc/) [9] using the GWAS summary statistics of the EURO dataset. European derived LD scores were used in LDSC (downloaded from https://doi.org/10.5281/zenodo.7768714).

We conducted a two-tailed t-test to compare the mean SNP-heritability between groups of phenotypes. The results were adjusted for multiple testing using the Benjamini-Hochberg procedure [63] (Fig B in S1 File).

### Identification of genetic loci

Peak calling was performed in three steps, starting with the SNPs that reached the adjusted genome-wide significance threshold ($5 \times 10^{-8}$ divided by the effective number of traits). First, all SNPs within ±250 kb of the most significant SNP, as well as those within 1 Mb and in LD ($r^2 > 10^{-2}$) were clumped into a single locus represented by the most significant (lead) SNP. This was repeated until all SNPs were assigned a locus. Next, any two loci were merged if the representative lead SNPs were within 10 Mb and in LD ($r^2 > 10^{-2}$). This locus was then represented by the SNP with the lowest P-value. Lastly, any peaks represented by a single SNP below the adjusted genome-wide significance threshold were disregarded to improve robustness.

### Gene annotation

The most likely candidate gene per lead SNP was identified through a two-step process. First, we utilized GREAT (v.4.0.4) [64] with default settings and the Table Browser of the UCSC Genome Browser [65] for gene annotation. Then, we conducted literature searches to further support our findings, based on the gene lists associated with facial morphology provided in [5].

### Supporting information

**S1 File.** Fig A. Boxplot of correlations between different groups of facial traits. Fig B. P-value matrix of pairwise differences in mean SNP-based heritability of different phenotyping categories. Fig C. Comparison of SNP-based heritability between phenotyping categories for nasal and facial shape. Fig D. Comparing facial phenotyping categories in terms of independent genetic loci identified in GWAS. Fig E. Frequency of genes identified by different categories of traits. Fig F. Using polynomial regression analyses to determine the optimal polynomial degree for each PC in predicting each dimension of AE. Fig G. LocusZoom plot for SNP rs1999464

based on different phenotyping methods.
(DOCX)

**S2 File.** Table A. Descriptive statistics SNP-based heritability of different phenotyping categories from facial shape GWASs. Table B. The number of overlapping genetic loci (within ±250kb) across all possible combinations of phenotype groups from nasal shape GWASs. Table C. The number of overlapping genetic loci (within ±250kb) across all possible combinations of phenotype groups from facial shape GWASs. Table D. Intersection of identified genes across all possible trait combinations from nasal shape GWASs. Table E. Intersection of identified genes across all possible trait combinations from facial shape GWASs. Table F. The number of overlapping genetic loci (within ±250kb) across all possible combinations of 10 replicates of AE-based phenotypes from nasal shape GWASs. Table G. The number of overlapping genetic loci (within ±250kb) across all possible combinations of 10 replicates for resemblance scores to randomly selected gestalts from nasal shape GWASs. Table H. The number of overlapping genetic loci (within ±250kb) across all possible combinations of 10 replicates of AE-based phenotypes from facial shape GWASs. Table I. The number of overlapping genetic loci (within ±250kb) across all possible combinations of 10 replicates for resemblance scores to randomly selected gestalts from facial shape GWASs. Table J. Annotated genes based on the frequently identified peaks from nasal shape GWASs. Table K. Annotated genes based on the frequently identified peaks from facial shape GWASs. Table L. Syndrome Data.
(XLSX)

**S1 Methods. Implementation details.**
(DOCX)

## Acknowledgments

We are extremely grateful to all the individuals and families who took part in this study, the midwives for their help in recruiting them and the whole teams at ALSPAC, KU Leuven, and the universities of Pittsburgh, IUI, and Penn State which includes interviewers, computer and laboratory technicians, clerical workers, research scientists, volunteers, managers, receptionists, and nurses.

We acknowledge the use of ChatGPT v3.5 (https://chat.openai.com/) for English language editing. More specifically, ChatGPT v3.5 was used to check English spelling and grammar, without changing meaning or adding content.

## Author Contributions

**Conceptualization:** Meng Yuan, Seppe Goovaerts, Peter Claes.

**Data curation:** Meng Yuan, Seppe Goovaerts, Michiel Vanneste, Harold Matthews, Hanne Hoskens, Stephen Richmond, Ophir D. Klein, Richard A. Spritz, Benedikt Hallgrimsson, Susan Walsh, Mark D. Shriver, John R. Shaffer, Seth M. Weinberg, Hilde Peeters, Peter Claes.

**Formal analysis:** Meng Yuan, Seppe Goovaerts.

**Funding acquisition:** Ophir D. Klein, Richard A. Spritz, Benedikt Hallgrimsson, John R. Shaffer, Seth M. Weinberg, Peter Claes.

**Investigation:** Meng Yuan, Seppe Goovaerts, Peter Claes.

**Methodology:** Meng Yuan, Seppe Goovaerts, Peter Claes.

**Project administration:** Peter Claes.

**Resources:** Michiel Vanneste, Harold Matthews, Hanne Hoskens, Stephen Richmond, Ophir D. Klein, Richard A. Spritz, Benedikt Hallgrimsson, Susan Walsh, Mark D. Shriver, John R. Shaffer, Seth M. Weinberg, Hilde Peeters, Peter Claes.

**Software:** Meng Yuan, Seppe Goovaerts, Harold Matthews.

**Supervision:** Hilde Peeters, Peter Claes.

**Visualization:** Meng Yuan.

**Writing – original draft:** Meng Yuan.

**Writing – review & editing:** Meng Yuan, Seppe Goovaerts, Michiel Vanneste, Harold Matthews, Hanne Hoskens, Stephen Richmond, Ophir D. Klein, Richard A. Spritz, Benedikt Hallgrimsson, Susan Walsh, Mark D. Shriver, John R. Shaffer, Seth M. Weinberg, Hilde Peeters, Peter Claes.

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
