## [Decision Letter · Decision Letter 0]

27 Jun 2024

Dear Ms Yuan,

Thank you very much for submitting your manuscript "Mapping genes for human face shape: exploration of univariate phenotyping strategies" for consideration at PLOS Computational Biology.

As with all papers reviewed by the journal, your manuscript was reviewed by members of the editorial board and by several independent reviewers. All three expressed enthusiasm of your work, but raised concerns, especially on the clarify of your presentation. In light of the reviews (below this email), we would like to invite the resubmission of a significantly-revised version that takes into account the reviewers' comments. 

We cannot make any decision about publication until we have seen the revised manuscript and your response to the reviewers' comments. Your revised manuscript is also likely to be sent to reviewers for further evaluation.

Sincerely,

Xin He

Guest Editor

PLOS Computational Biology

Ilya Ioshikhes

Section Editor

PLOS Computational Biology

Dear Dr. Yuan,

Your manuscript, "Mapping genes for human face shape: exploration of univariate phenotyping strategies", has been reviewed by three referees. All three showed enthusiasm of your work, but raised concerns, especially on the clarify of your presentation. Please revise your manuscript accordingly, and we will be happy to review the revised version.

Best,

Dr. Xin He

Reviewer's Responses to Questions

**Comments to the Authors:**

Reviewer #1: Please see attached reviewer comment file.

Reviewer #2: In this work, the authors have compared three different approaches to extract facial features as a trait for univariate GWAS. The result shows that inter-landmark has higher power and provide higher heritability estimate in general. It is interesting that the three approaches are complimentary in terms of genetics loci discovery. I only have two minor comments:

(1) Minimum P values was taken and then the Bonferroni method was applied to correct multiple testing across traits. This may be too stringent though Fig 3 suggest adding more traits still improved power. Authors may try to apply meta analysis to aggregate multiple traits within the same approach as mentioned in the introduction. Alternatively, the Sime's method may be more suitable for multiple testing correction across correlated traits.

(2) Sharing of genomic signals were identified with a distance threshold of 250kb. This seems arbitrary. Please also show how many siginificant loci are overlapped without extending 250kb. Also it would be helpful to show the Manhattan plots from different methods in parrallel vertically so readers can check the sharing more closely. If possible, please try to use MASH (PMID: 30478440) to analyze the sharing of genetic effects between different phenotyping methods.

Reviewer #3: In this paper, Meng et al. set out to explore how the choice of different face modeling approaches can impact on GWAS findings. They compares four different strategies of modeling facial features, include pairwise landmark distances, two unsupervised dimension reduction techniques (PCA and AE), and three different facial scores that based on supervised learning. The metrics of the comparisons are SNP heritability estimates, significant GWAS hits, and the overlaps with GWAS hits reported other than face GWAS. I find this topic very interesting. The analytic plans are comprehensive and relevant. The results in general are very informative to the researchers who are interested in modeling the morphological traits and finding their corresponding genetic architectures. However, because the complexity of such approach, the clarity of the manuscript need to be improved in order to reach wider audiences.

1. It is unclear how exactly many features were used in each analytic procedures. The numbers were scattered across manuscript and have to fish them out. Those numbers are critical, as the number of feature tested can have differential impact on the number of GWAS hits found. Larger the number, higher the chance to find high heritability and GWAS hits.

2. It is very nice that the author attempted to address the issue of different number of tests involved in each different method, as shown in the Figure 3. But exactly because of the effective number of features are different across methods, the conclusion based on the distribution of the GWAS findings is weakened. As shown in the Figure 3B and 3C, the asymptotic behaviors have not been kicked in before the number of features are exhausted.

3. I found that the Fig4 is very confusing. It is hard to trace all the different numbers across a large matrices. It would be better if they can summarize the results with proportion instead of absolute counts. However, I acknowledge the difficulties in doing as, as the absolute raw count is already low.

4. The resemblence score approach is very interesting and has higher yield of GWAS hits despite low heritability. However, I am not sure what exactly those scores are trying to capture, especially those modeled with "extreme" cases and "syndrome" groups. It also makes me wonder what is exactly the number of effective features mean here, since there can be large number of "extreme" or "syndrome" groups that they have not modeled yet.

**Have the authors made all data and (if applicable) computational code underlying the findings in their manuscript fully available?**

Reviewer #1: Yes

Reviewer #2: Yes

Reviewer #3: Yes

PLOS authors have the option to publish the peer review history of their article (what does this mean?). If published, this will include your full peer review and any attached files.

Reviewer #1: No

Reviewer #2: No

Reviewer #3: **Yes: **Chun Chieh Fan
---

## [Decision Letter · Decision Letter 1]

11 Oct 2024

Dear Ms Yuan,

Thank you very much for submitting your manuscript "Mapping genes for human face shape: exploration of univariate phenotyping strategies" for consideration at PLOS Computational Biology. As with all papers reviewed by the journal, your manuscript was reviewed by members of the editorial board and by several independent reviewers. The reviewers appreciated the attention to an important topic. Based on the reviews, we are likely to accept this manuscript for publication, providing that you modify the manuscript according to the review recommendations.

The reviewers feel most of the comments have been addressed by this revision. Nevertheless, there is a remaining concern:

In Figure 4 and Table S2-S5, the authors used the UpSet plot to represent the overlap of GWAS variants across different methods. It is good to classify them in terms of methods, however, this UpSet plot only showed the unique identifications and pairwise identifications. It would be beneficial to have an UpSet plot including all possible sharing patterns and also ranked by inclusive intersections. I understand the number of all possible sharing patterns should be a huge number, the authors could show the top 20-25 patterns. It could be more clear to show the relationship across different methods. Additionally, I have a follow up question about the consistency of the AE method. The authors replicated the AE method three times (AE1, AE2, AE3), but based on Table S2-S5, these three AE replicates identify different variants (overlap variants are few). It is better to have a discussion about the consistency of the AE method.

Sincerely,

Xin He

Guest Editor

PLOS Computational Biology

Ilya Ioshikhes

Section Editor

PLOS Computational Biology

The reviewers feel most of the comments have been addressed by this revision. Nevertheless, there is a remaining concern:

In Figure 4 and Table S2-S5, the authors used the UpSet plot to represent the overlap of GWAS variants across different methods. It is good to classify them in terms of methods, however, this UpSet plot only showed the unique identifications and pairwise identifications. It would be beneficial to have an UpSet plot including all possible sharing patterns and also ranked by inclusive intersections. I understand the number of all possible sharing patterns should be a huge number, the authors could show the top 20-25 patterns. It could be more clear to show the relationship across different methods. Additionally, I have a follow up question about the consistency of the AE method. The authors replicated the AE method three times (AE1, AE2, AE3), but based on Table S2-S5, these three AE replicates identify different variants (overlap variants are few). It is better to have a discussion about the consistency of the AE method.

Reviewer's Responses to Questions

**Comments to the Authors:**

Reviewer #1: The reviewer addressed most of my comments. I have a minor suggestion regarding the current Figure 4, the UpSet plot. It would be better to categorize and rank the data into different phenotypic categories, similar to the organization in your previous Figure 4. For example, the top category could be DISTANCE, followed by the three AE replicates (AE1, AE2, AE3), and then the three RANDOM replicates (RANDOM1, RANDOM2, RANDOM3). This approach would help readers understand the similarities across the three AE replicates and the pairwise comparisons among them.

Reviewer #2: The authors have addressed all my concerns.

**Have the authors made all data and (if applicable) computational code underlying the findings in their manuscript fully available?**

Reviewer #1: Yes

Reviewer #2: Yes

PLOS authors have the option to publish the peer review history of their article (what does this mean?). If published, this will include your full peer review and any attached files.

Reviewer #1: No

Reviewer #2: No

Figure Files:

Data Requirements:

Reproducibility:

References:

*If you need to cite a retracted article*, *indicate the article’s retracted status in the References list and also include a citation and full reference for the retraction notice.*

---

## [Editor Report · Decision Letter 2]

5 Nov 2024

Dear Ms Yuan,

We are pleased to inform you that your manuscript 'Mapping genes for human face shape: exploration of univariate phenotyping strategies' has been provisionally accepted for publication in PLOS Computational Biology.

Best regards,

Xin He

Guest Editor

PLOS Computational Biology

Ilya Ioshikhes

Section Editor

PLOS Computational Biology

Feilim Mac Gabhann

Editor-in-Chief

PLOS Computational Biology

Jason Papin

Editor-in-Chief

PLOS Computational Biology

The reviewer feels that the remaining comment has been adequately addressed. Thus I would recommend to accept the paper.

---

## [Editor Report · Acceptance letter]

21 Nov 2024

PCOMPBIOL-D-24-00574R2 

Mapping genes for human face shape: exploration of univariate phenotyping strategies

Dear Dr Yuan,

I am pleased to inform you that your manuscript has been formally accepted for publication in PLOS Computational Biology. Your manuscript is now with our production department and you will be notified of the publication date in due course.

With kind regards,

Anita Estes
